# Empirical distributions of time intervals between COVID-19 cases and more severe outcomes in Scotland

**Anthony J. Wood**[1], **Rowland R. Kao**[1,2]*

**1** The Roslin Institute, University of Edinburgh, Edinburgh, United Kingdom, **2** Royal (Dick) School of Veterinary Studies, University of Edinburgh, Edinburgh, United Kingdom

* Rowland.Kao@ed.ac.uk

## Abstract

A critical factor in infectious disease control is the risk of an outbreak overwhelming local healthcare capacity. The overall demand on healthcare services will depend on disease severity, but the precise timing and size of peak demand also depends on the time *interval* (or clinical time delay) between initial infection, and development of severe disease. A broader *distribution* of intervals may draw that demand out over a longer period, but have a lower peak demand. These interval distributions are therefore important in modelling trajectories of e.g. hospital admissions, given a trajectory of incidence. Conversely, as testing rates decline, an incidence trajectory may need to be inferred through the delayed, but relatively unbiased signal of hospital admissions. Healthcare demand has been extensively modelled during the COVID-19 pandemic, where localised waves of infection have imposed severe stresses on healthcare services. While the initial acute threat posed by this disease has since subsided with immunity buildup from vaccination and prior infection, prevalence remains high and waning immunity may lead to substantial pressures for years to come. In this work, then, we present a set of interval distributions, for COVID-19 cases and subsequent severe outcomes; hospital admission, ICU admission, and death. These may be used to model more realistic scenarios of hospital admissions and occupancy, given a trajectory of infections or cases. We present a method for obtaining empirical distributions using COVID-19 outcomes data from Scotland between September 2020 and January 2022 ($N = 31724$ hospital admissions, $N = 3514$ ICU admissions, $N = 8306$ mortalities). We present separate distributions for individual age, sex, and deprivation of residing community. While the risk of severe disease following COVID-19 infection is substantially higher for the elderly and those residing in areas of high deprivation, the length of stay shows no strong dependence, suggesting that severe outcomes are equally severe across risk groups. As Scotland and other countries move into a phase where testing is no longer abundant, these intervals may be of use for retrospective modelling of patterns of infection, given data on severe outcomes.

**Data Availability Statement:** The outcomes data utilised in this work are not publicly available. They are provided to the authors for academic research by Public Health Scotland's electronic Data

Research and Innovation Service, under a data
sharing agreement (Spatial and Network Analysis
of SARS-CoV-2 Sequences to Inform COVID-19
Control in Scotland) and be contacted via phs.
edris@phs.scot. We received no special privileges
with respect to data access as compared to other
researchers. Deprivation data are obtained from the
2020 Scottish Index of Multiple Deprivation, which
is publicly available (url https://www.simd.scot).
Nationwide-level occupancy of patients with
COVID-19 are obtained from Public Health
Scotland, and is also publicly available (url https://
www.gov.scot/publications/coronavirus-covid-19-
trends-in-daily-data/).

**Funding:** This study was funded by the Economic
and Social Research Council (grant number ES/
W001489/1) awarded to RRK. This study was also
funded by the Roslin ISP2 (theme 3) (grant
number BBS/E/D/20002174) (https://gow.bbsrc.
ukri.org/grants/AwardDetails.
aspxFundingReference=BBS/E/D/20002174) for
which RRK is a Co-Investigator. The sponsor had
no role in the design, data collection, analysis,
decision to publish or preparation of this
manuscript.

**Competing interests:** The authors have declared
that no competing interests exist.

## Introduction

The threat posed by an infectious disease on a human population depends critically on the risk of developing severe illness, and the burden that may place on healthcare systems. During the early stages of transmission or a new wave of infection, it is important to understand when and how to expect that future demand on healthcare systems to come. Depending on the natural history of the disease, this may be immediate and acute, or be drawn out over a longer period of time.

Such estimation of future healthcare burden has been particularly important during the COVID-19 pandemic, where severe curbs on everyday life were imposed globally, in order to control the spread of infection and prevent the overwhelming of hospital capacity. Now over three years from the first known case, the acute threat posed by the virus has subsided, with the build-up of population immunity via effective vaccination and prior infection. However, future pressures on healthcare services due to COVID-19 may still be considerable, with a continuing circulation of infection and waning immunity on timescales of order six months [1].

In prior waves of COVID-19 infection, the peak demand on healthcare services has followed, with a delay, a peak in incidence. Our work here focuses on the length of this delay and its dependence on age, sex, and local deprivation; risk factors that are known to be important for determining the likelihood of severe infection. For modelling future waves of infection, the delay between infection and developing of severe disease (and variation in that delay from person to person) introduces uncertainty on the timing and size of the future *peak* demand on healthcare services, even if an overall infection-hospitalisation rate is well estimated.

The COVID-19 pandemic has been unique in the volume and granularity of data recorded for a human disease. This is not just for hospital admissions and deaths (for which data are already routinely collected for many illnesses), but on proactive testing for the disease. Using such data, the aim of this work is to then obtain empirical distributions of the time intervals between different outcomes at the individual level. We use data collected across Scotland, between September 2020 and January 2022. In this period, COVID-19 tests were widely and freely available, with reported results collected in central databases.

The Scottish COVID-19 data are advantageous for our study as they include additional identifiers in the data, allowing us to show how these distributions differ by a person's age, sex, and deprivation in their residing community, all of which are known risk factors for poor COVID-19 outcomes [2–7]. Our distributions build on length-of-stay distributions obtained from the first year of the pandemic [8–14], as well as inferred distributions of intervals from the initial infection stage [15], and from onset of symptoms to diagnosis or mortality [16].

The first set of distributions we describe are between COVID-19 cases, and three different severe outcomes: hospital admission, intensive care unit (ICU) admission, and mortality. These affect the shape of the trajectory of severe outcomes, given a trajectory of cases. Interval distributions with a higher mean value will lead to a greater delay between cases and following severe outcomes, whereas a higher variance will lead to a more drawn-out trajectory, with a lower peak.

The second set of distributions are those *between* different severe outcomes. These relate to hospital occupancy; how long an individual spends in hospital once admitted. While the data do not specify recoveries or discharges, we also estimate the distribution of intervals between admission and discharge, for patients admitted that go on to *survive*, using public data on hospital occupancy. We infer that over time, an increasing proportion of COVID-19 hospital burden in the period studied was of individuals that eventually went on to be discharged, and the mean time spent in hospital shortened.

## Materials and methods

### Empirical distributions

COVID-19 data are provided by Public Health Scotland's (PHS) *electronic Data Research and Innovation Service* (eDRIS). These data were accessed under a COVID-19 rapid response application to the Public Benefits and Privacy Panel (see [17]). Under this initial rapid response, no additional ethics approval was required, however ethical approval was later retrospectively obtained from the Edinburgh University College of Medicine and Veterinary Medicine Research Ethics and Integrity Assessment Panel. All data under this request were originally collected by PHS for the purposes of monitoring the progression of COVID-19 in Scotland. As such, the requirements for notification and conditions for consent are as found at [18].

The data specify COVID-19 tests and severe outcomes. Each entry has an associated date, and de-identified patient ID with age (in five-year windows, up to 75+), sex, and residing *data zone* (DZ). DZs are non-overlapping Scottish census areas, each with a residing population of order 500–1,000. There are 6,976 DZs in total, covering the full area and population of Scotland. A subpopulation of a certain DZ, age range, and sex (e.g., men aged between 50–54 living in a particular DZ) will typically have 0–50 individuals, allowing us to analyse outcomes at this same resolution.

The test data contain results from both rapid lateral flow device (LFD) tests and polymerase chain reaction (PCR) tests. In the period studied, public health policy was that those exhibiting COVID-19 symptoms or testing positive on an LFD should report the result, and take a confirmatory PCR test (usually at a local testing centre). We define a case as a new positive test result, taken at least 60 days from any previous positive test by the same individual. For those with repeat positive tests within 60 days (such as a LFD positive followed by a PCR positive), we use the date of the first PCR positive, or the first LFD positive if there is no PCR positive. The case date is the date on which the test is taken.

A *severe* COVID-19 outcome—a hospital admission, intensive care unit (ICU) admission, or mortality—is one where "*coronavirus*" or "*COVID-19*" appears within the underlying causes for that outcome. This analysis therefore includes severe outcomes "with" COVID-19. The date associated with hospitalisation and ICU admission is the date of first admission, and the date associated with mortality is the day the individual dies. We assume that if an individual dies after a hospitalisation or ICU admission, then they died in hospital, and were in hospital for the whole time.

Outcomes between September 10 2020 and January 6 2022 were considered. In this period of the epidemic testing was both free, and widely available to the whole population in Scotland. We exclude data for those aged under 20, owing to a scarcity of severe outcomes. Presented values draw from the version of the eDRIS data dated October 27 2022.

Data on overall COVID-19 hospital occupancy are available publicly from PHS, with daily figures on patients in hospital with confirmed COVID-19, throughout the period studied [19].

In our analysis we define an *interval* $\Delta t_{AB}$ as the elapsed time between two different outcomes A and B. This is given as a whole number of days. We obtain distributions for:

- *Case* intervals, between cases and more severe outcomes: Case-to-Hospitalisation ($\Delta t_{CH}$), Case-to-ICU admission ($\Delta t_{CI}$), Case-to-Mortality ($\Delta t_{CM}$), and;

- *Nosocomial* intervals, between different outcomes while in hospital, affecting length of stay: Hospitalisation-to-ICU admission ($\Delta t_{HI}$), Hospitalisation-to-Mortality ($\Delta t_{HM}$), ICU admission-to-Mortality ($\Delta t_{IM}$).

We present further details on how the high resolution of the data allows different outcomes to be associated in S1A Appendix.

Alongside the empirical distributions, we fit the distributions using gamma curves, with shape $\alpha$ and rate $\beta$. For the case intervals, we include an additional parameter $\nu$ to quantify the proportion of events where the interval was zero days. Further details of our fitting methodology are presented in S1B Appendix.

### Estimation of hospitalisation-to-discharge intervals

The eDRIS data do not specify date of recovery or discharge, for individuals admitted to hospital with COVID-19 but later discharged (and assumed to recover). Using a standard approximate Bayesian computation (ABC) algorithm, we estimate a distribution $P(\Delta t_{\mathrm{HD}})$ for the nosocomial *Hospitalisation-to-Discharge* intervals $\Delta t_{\mathrm{HD}}$ for admissions that do not have an associated death, by comparing to the trajectory of hospital *occupancy*, for those with confirmed COVID-19 [19].

In order to do this we first infer the trajectory of admissions of patients that specifically go on to survive, by identifying those that do *not* have an associated mortality. We then take the difference between nationwide hospital occupancy of all COVID-19 patients and the known occupancy of patients that die (using the hospitalisation-to-mortality intervals), to infer the occupancy of patients that go on to survive. We then fit a simple exponential distribution of hospitalisation-to-discharge intervals that best reproduce this occupancy. Finally, to account for changes in the epidemic during the period studied, we consider two separate periods, first with data up to April 30 2021 preceding the first major wave of the *Delta* variant termed the "pre-Delta" period, and data from May 1 2021 onwards the "post-Delta" period. Additional details of our inference are presented in in S1C Appendix.

## Results

### Interval distributions

Fig 1 summarises all empirical interval distributions.

For outcomes where a corresponding case was found, we first note that for hospital admissions and ICU admissions, a significant proportion (28% and 11% respectively) of these were dated the same day as the case. This proportion remains broadly consistent throughout the period studied. *Excluding* these same-day events, the mean case-to-hospitalisation interval ($\Delta t_{\mathrm{CH}}$) was 6.95 days (standard deviation (s.d.) 4.39 days). Similarly, the mean case-to-ICU admission interval was 7.31 days (s.d. 4.17 days) and the mean case-to-mortality interval was 11.94 days (s.d. 6.54 days). 24% of hospital admissions (7,580/31,724), 11% of ICU admissions (398/3,514) and 17% of mortalities (1,399/8,306) had no associated case. These "no-case" entries likely include instances where the individual age/sex/DZ were inconsistent across the data, as well as instances where COVID-19 was identified as a cause, but without a positive test being reported on or before admission.

When considering age, case intervals contract slightly in older age groups with the mean case-to-hospitalisation interval falling from 7.41 days to 6.48 days from ages 20–49 to 70+, whereas the standard deviation remains of order 4–5 days (S1 and S2 Tables and S1 Fig in S1 File). A larger discrepancy is seen in same-day admissions (amongst those with an associated case at all), with 39% (3,814/9,743) of hospital admissions amongst those aged 70+ found on the same day as testing, falling to 16% (936/5,866) for those aged 20–49.

There is a smaller difference considering the level of local deprivation, with the proportion of same-day admissions increasing from 25% (1,374/5,439) in the least deprived third of DZs,

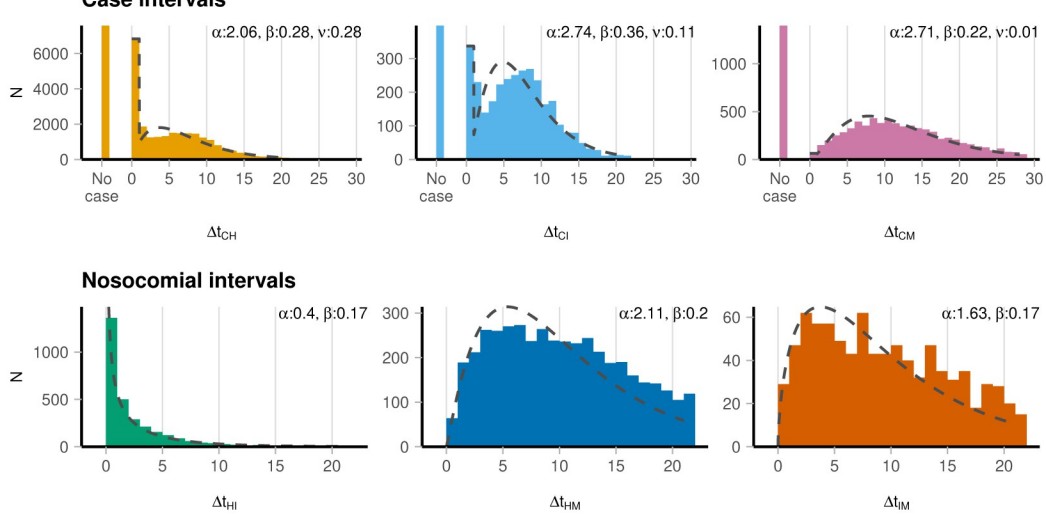

**Fig 1. Interval distributions across all ages 20+, comparing time between a (C)ase, (H)osptial admission, (I)CU admission, and (M)ortality.** The "no case" entries represent instances where no associated case was found for that severe outcome. Fit parameters are for gamma distributions where $\alpha$ is the shape, $\beta$ the rate, and $\nu$ a zero-inflation factor for case intervals).

to 30% (3,484/11,560) in the most deprived third. By sex, 27% (3,237/11,878) of admissions amongst women were on the same day, compared to 29% (3,600/12,266) for men.

Fig 1 also summarises nosocomial intervals (with full summary statistics in S2 Table), relating to hospital time of stay. The overall mean interval from hospitalisation to ICU admission $\Delta t_{HI}$ was 2.13 days (s.d. 3.34 days), this time with younger age groups having shorter intervals. The mean hospitalisation-to-mortality interval $\Delta t_{HM}$ was 9.61 days (s.d. 5.65 days). For those that died after admission to an ICU, the mean interval $\Delta t_{IM}$ was 8.90 days (s.d. 5.80 days).

## Variability in outcomes across different timeframes

Between September 2020 and January 2022, the epidemic in Scotland evolved in several characteristic ways. First, the dominant coronavirus variant switched three times (the initial *wild* type was followed by the *Alpha* variant introduced around November 2020, followed by the *Delta* variant introduced around May 2021, finally being replaced by the *Omicron* (BA.1) lineage, introduced around November 2021). Additionally, Scotland's COVID-19 vaccination programme began in December 2020, and by January 1 2022 over 11 million doses had been administered to a population of 5.5 million, with high uptake across all age groups [20]. This was in addition to the introduction of novel antiviral treatments, available to the most vulnerable [21]. The corresponding change in *intervals* throughout the period are then presented in Fig 2. Case intervals remain broadly consistent (as do the proportion of "same-day" admissions, see S1 File (Fig 3b)). $\Delta t_{IM}$ falls slightly. While the period in and around June 2021 sees some large fluctuations in the rolling mean (particularly in $\Delta t_{IM}$ and $\Delta t_{CM}$), far fewer severe outcomes were recorded and trends in this period have much greater uncertainty (as illustrated with a larger standard error on the mean).

## Estimated hospitalisation-to-discharge interval

In the "pre-Delta" period, the mean interval (across all accepted $P(\Delta t_{HD})$) was $E(\Delta t_{HD}) \sim 13.2 \pm 0.8$ days, falling to $E(\Delta t_{HD}) \sim 9.5 \pm 0.5$ days in the "post-Delta" period. The

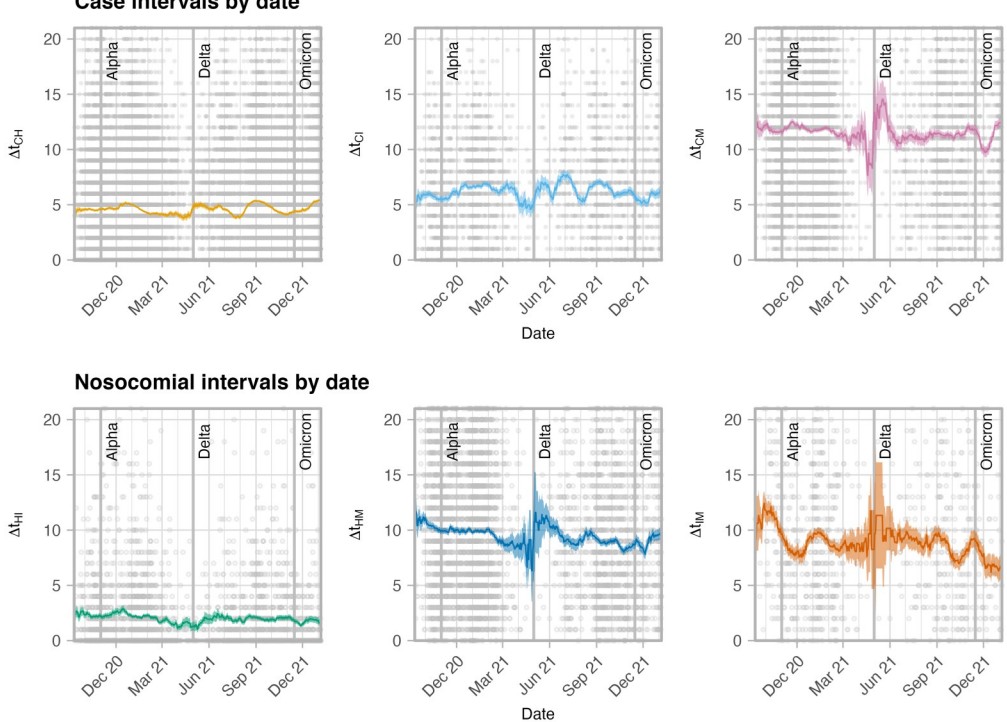

**Fig 2. Time evolution of the interval distribution, with points indicating individual instances (taking date of the first outcome, e.g. a case-to-hospitalisation interval is plotted on the date of the case), and the line representing the rolling mean of intervals in a 28-day window centered around that date, and the filled region representing the standard error of the rolling mean, indicating a higher error in periods with fewer data (points).** The approximate times of introduction of the Alpha, Delta and Omicron variants into Scotland (November 1 2020, May 1 2021 and November 15 2021 respectively) are marked with vertical lines.

corresponding posterior distributions (for a zero-inflated exponential distribution with rate $\beta$ and inflation $\nu$) are given in Fig A in S1 Appendix.

Between September 10 2021 and January 6 2022, PHS reported a total of 38,480 admissions, whereas 32,718 were derived from the eDRIS data in the same period. With fewer admissions, we may then over-estimate the mean interval of each stay. There is also a clear time dependence; the admissions trajectory sees a peak of 241 on January 11 2021, 11 days prior to an occupation peak of 2,053. A later admissions peak of 202 on September 7 2021 then precedes a much lower occupation peak of 1,107, 14 days later, suggesting those admitted were staying for less time (see S1 File (Fig 2a, 2b)). This motivates the choice to separately fit two periods. From December 2021 onward, hospital admissions are most likely attributable to infection with the *Omicron* variant, where material changes in admissions trends would be expected as compared to previous variants, owing to a lower observed severity [22].

The data also reveal that in this period, $\sim$80–90% of both admissions and hospital occupancy were from individuals that were eventually discharged (and assumed survived) (S1 File (Fig 2c–2h)). The occupancy proportion is a distinct metric from hospitalisation survival *rates*, as it may also change if the mean length of hospitalisation stay by outcome changed substantially.

## Discussion

The COVID-19 pandemic has seen an unprecedented level of consistent, voluntary, recorded testing, in effort to detect and control the spread of infection. This volume of testing, in

combination with precise data routinely collected on hospitalisation and mortality, allows us to analyse the natural history of COVID-19 disease at the level of the individual. The monitoring of other infectious diseases in comparison is usually far more rudimentary; the endemic prevalence of influenza in the UK for example is regularly estimated by the volume of phone calls to GPs citing influenza-like symptoms [23]. With the volume of recorded COVID-19 testing since falling to a fraction of that in the period studied here, this period is a unique window for studying intervals of an infectious disease at such detail.

Our work here uses COVID-19 outcomes data in Scotland to draw empirical distributions of intervals between cases and subsequent severe outcomes, as well as intervals between those severe outcomes (e.g., the time between hospital admission and death). The case-to-hospitalisation intervals (Fig 1) in particular inform, given a surge in COVID-19 cases, how delayed (dependent on the mean value) and drawn out (dependent on the variance) a corresponding trajectory in hospital admissions may be. This is important for estimating the timing and size of peak demand on healthcare services (and whether that could exceed capacity).

These intervals have a clear age structure. Differences are mainly seen in the proportion of "same-day" events; 16% of all age 20–49 admissions were on the same day compared to 39% for ages 70+, whereas the distribution for those not admitted on the same day only changed modestly. This larger same-day variation is also seen with respect to deprivation. This suggests a difference in risk profiles between those testing prior to admission (where severe symptoms develop later), and those testing on the same day or on admission. A likely influencing factor is admissions from those admitted to hospital for a non-COVID-19 reason, but testing positive on a routine test on admission. Prior to the introduction of the Omicron variant the proportion of COVID-19 admissions being "with" COVID-19 remained at approximately 25% (S1 File) (Fig 3a)).

As well as age, deprivation and sex have been highlighted as risk factors for poor COVID-19 outcomes, with higher rates associated with men [24–26], and individuals living in more deprived communities [2, 27, 28]. Our analysis reaffirms this in the overall number of severe outcomes per group, and reveal modest variation in their intervals (S1 and S2 Tables), with a slightly higher proportion of same-day admissions from both men, and individuals living in more deprived communities.

Compared to cases (dated to when a test is *taken*), the delay from *infection* or *onset of symptoms* to these severe outcomes will be longer, taking into account the incubation period of the disease (estimated to have a median of order five days [29, 30]), and a potential delay between onset of symptoms and taking a test. In light of this, our results are then broadly consistent with intervals presented by a UK study [15], which infers the mean interval from infection to hospitalisation prior to January 2021 to have been of order 8–10 days (as compared with our mean case-to-hospitalisation interval of 6.95 days), and time from infection to mortality to have been of order 9–16 days (compared to our mean case-to-mortality interval of 11.94 days). A more recent work in [16] studies intervals in South Korea over a similar period to that studied here, and finds a longer mean interval from symptom onset to death of 20.1 days, and from symptom *reporting* to death of 16.7 days. The measured interval from cases or onset to more severe outcomes will have several influencing factors. These include but are not limited to the frequency of testing (in turn the mean time after infection that a test is taken), the properties of the virus and differences across different variants and, as highlighted here, the structure of the population with respect to age. It is therefore reasonable to expect variation in absolute values, but trends such as a shortening in intervals with increasing age appear to be consistent across studies [15, 16].

While the *nosocomial* intervals—relating to time of stay in hospital—are routinely collected, they are not often available to researchers at such detail, with the data used here an exception

owing to the need for modelling support in the acute stage of this pandemic. We see variation in these intervals through the period studied. For those admitted to hospital that go on to die, the interval from admission to death shortens over the period (Fig 2). This is somewhat counter-intuitive given improving rates of survival after admission (S1 File (Fig 2g, 2h)), but may be due to individuals who recover and are discharged, that may have died if infected at an earlier stage of the pandemic (where treatment of severe COVID-19 disease was less well understood). If one then assumes that these "removed' individuals are generally healthier and would have stayed alive in hospital for longer, the removal of these individuals would reduce the mean interval overall.

Finally, once same-day events are excluded we see little between-group variation in the standard deviation of the intervals. This variation influences how drawn out a trajectory of severe outcomes may be relative to that of cases (or hospital admission/ICU admission for nosocomial intervals). The lack of variation then suggests that the shape of the trajectory is not strongly dependent on whom amongst the population has been infected (though the volume, of course, certainly will).

While the interval distributions are drawn from a comprehensive national-scale dataset, our study does have some limitations. The first is an inevitable selection bias whereby the case intervals are determined from individuals that volunteered to test. Further, while PCR testing was generally freely available for those exhibiting symptoms throughout the period, rapid LFD testing (where frequent testing may detect a COVID-19 infection earlier) was only available from April 2021 onwards [31], which may have affected the interval. In addition, the outcomes data analysed were gathered whilst the first nationwide COVID-19 vaccination programme was conducted, and our distributions do not distinguish between individuals that were or were not vaccinated. We also do not differentiate between scenarios where COVID-19 was a primary or secondary diagnosis, when approximately a quarter of COVID-19 admissions in the data were indeed secondary.

The programme of free community testing in Scotland has allowed the spread of the virus to be tracked at remarkable spatio-temporal resolution [2, 32–37]. Despite prevalence remaining high throughout 2022 and early 2023 per randomised testing [38], Scotland has since entered a phase where testing is no longer mandatory, nor generally free of charge. The proportion of COVID-19 infections being identified is now very low. The absence of such testing data may result in future outbreak modelling relying on the trajectories of severe outcomes only, or more basic estimates of incidence. Combined with a reasonable understanding of the incubation period and potential case ascertainment, our interval distributions may help infer routes of transmission and patterns of infection.

## Supporting information

**S1 Table. Summary statistics of case intervals.** These are specified by age, sex and the relative deprivation of residing DZ (equally dividing the DZs into three groups, based on the overall rank in the *Scottish Index of Multiple Deprivation*).
(PDF)

**S2 Table. Summary statistics of nosocomial intervals.** These are specified by the same groups as the case intervals.
(PDF)

**S1 File.**
(PDF)

**S1 Appendix.**
(PDF)

## Acknowledgments

We thank eDRIS for the provision of COVID-19 testing and severe outcomes data. We also thank the reviewers for their helpful feedback and suggestions, which has led to improvement of the manuscript.

## Author Contributions

**Conceptualization:** Rowland R. Kao.

**Formal analysis:** Anthony J. Wood.

**Funding acquisition:** Rowland R. Kao.

**Investigation:** Anthony J. Wood.

**Methodology:** Anthony J. Wood, Rowland R. Kao.

**Supervision:** Rowland R. Kao.

**Writing – original draft:** Anthony J. Wood, Rowland R. Kao.

**Writing – review & editing:** Anthony J. Wood, Rowland R. Kao.

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
