## [Decision Letter · Decision Letter 0]

4 Apr 2023

PONE-D-22-32220Time intervals between COVID-19 cases and more severe outcomesPLOS ONE

Dear Dr. Kao,

Thank you for submitting your manuscript to PLOS ONE. The reviewers have pointed out a few aspects that could be improved and suggested a minor revision. Therefore, we invite you to submit a revised version of the manuscript that addresses the points raised during the review process.

 Please submit your revised manuscript by May 19 2023 11:59PM. If you will need more time than this to complete your revisions, please reply to this message or contact the journal office at plosone@plos.org. Please include the following items when submitting your revised manuscript:A rebuttal letter that responds to each point raised by the academic editor and reviewer(s). You should upload this letter as a separate file labeled 'Response to Reviewers'.A marked-up copy of your manuscript that highlights changes made to the original version. You should upload this as a separate file labeled 'Revised Manuscript with Track Changes'.An unmarked version of your revised paper without tracked changes. You should upload this as a separate file labeled 'Manuscript'.We look forward to receiving your revised manuscript.

Kind regards,

Alberto Aleta

Academic Editor

PLOS ONE

Journal Requirements:

5. We note you have included a table to which you do not refer in the text of your manuscript. Please ensure that you refer to Table 1 and 2 in your text; if accepted, production will need this reference to link the reader to the Table.

Reviewers' comments:

Reviewer's Responses to Questions

**Comments to the Author**

1. Is the manuscript technically sound, and do the data support the conclusions?

Reviewer #1: Yes

Reviewer #2: Yes

2. Has the statistical analysis been performed appropriately and rigorously? 

Reviewer #1: Yes

Reviewer #2: Yes

3. Have the authors made all data underlying the findings in their manuscript fully available?

Reviewer #1: Yes

Reviewer #2: Yes

4. Is the manuscript presented in an intelligible fashion and written in standard English?

Reviewer #1: Yes

Reviewer #2: Yes

5. Review Comments to the Author

Reviewer #1: This paper presents an analysis of Covid-19-related hospitalization data in Scotland, between September 2020 and January 2022. Two different data sets are used and combined to estimate the distributions of time between given events in disease history: positive test, hospitalization, ICU admission and death. Anonymized patient-level data including age, sex and deprivation level of the area of residence were obtained from the Scottish electronic database.

Overall, this paper presents interesting results using data collected in a period of intense scrutiny, which will be useful in the future to fit models for Covid-19 outbreaks in regions that are less well-sampled. I believe that this paper can be published in Plos ONE. Below are some minor comments that might improve the presentation.

1. In the introduction, you mention variance of the interval distributions as a key feature related to the way in which hospital occupancy peaks are drawn out over time with respect to incidence peaks. This is important and justifies the fact that you are fitting distributions instead of comparing mean (or median) times. Perhaps you could add the variance estimates to Tables 1 and 2, as well as a brief discussion of variance in the Results or Discussion sections? The confidence intervals are not very informative in this case.

2. In a similar vein, confidence intervals could be added to the graphs in Fig. 2: one would assume that in the post-vaccination lull in the summer of 2021, low patient numbers lead to higher estimation error, which could explain the sudden jumps in t_CM and t_HM with the arrival of the Delta variant (or maybe this is a genuine effect related to epidemic dynamics?) In any case, it could be explored further, especially if these methods are to be applied to future real-time data, to prevent such artifacts from being mistaken as real shifts in the underlying interval distribution.

3. Regarding the distribution fits of Fig. 1, it would be useful to add a few words on the method used. Also, why was a Gamma distribution chosen over e.g. a lognormal distribution?

4. Some typos:

- p. 5: define the abbreviation "DZ" as it's the first time it is used in the text.

- p.6, Fig.1: "(H)ospial" should be "(H)ospital"

- p.17, "If at least one one possible..." should be "If at least one possible..."

Reviewer #2: The authors accessed outcome data from Public Health Scotland and collected publicly available data of deprivation data and nationwide-level occupancy of patients with COVID-19. They aimed to present a method for obtaining empirical distributions using COVID-19 outcomes data from Scotland between September 2020 and January 2022.

The authors report separate distributions for individual age, sex, and deprivation of residing community. They observed that the risk of severe disease following COVID-19 is higher for the elderly or those who reside in areas of high deprivation, the length of stay shows no strong dependence, suggesting that severe outcomes are equally severe across the risk groups. This study is likely to contribute to public health decision-making (or say will support future analysis of the disease, such as modelling and simulation). It is, therefore, worthy of publication, and the analysis is reasonably well-supported. However, I find some minor points that need to be addressed before publication can occur.

Minor comments:

- The Appendix D (Estimation of hospitalisation-to-discharge intervals) seem better suited for the main text methods, contingent on the editor's approval.

- I recommend changing the colour of the figures to a colour-blind-friendly palette.

- I suggest that the authors change the manuscript title to describe the study's locality.

- Better present tables 1 and 2. The captions of both tables must be improved. “Case intervals” and “Nosocomical intervals” are not entirely describing what is in the tables.

- It Would be good to describe the study limitations, if any.

6. PLOS authors have the option to publish the peer review history of their article (what does this mean?). If published, this will include your full peer review and any attached files.

Reviewer #1: No

Reviewer #2: **Yes: **Cleber Vinicius Brito dos Santos

---

## [Author Response · Author response to Decision Letter 0]

10 May 2023

We have enclosed our item-by-item response to the reviewer comments in ReviewerLetter.pdf.

---

## [Editor Report · Decision Letter 1]

5 Jun 2023

Empirical distributions of time intervals between COVID-19 cases and more severe outcomes in Scotland

PONE-D-22-32220R1

Dear Dr. Kao,

We’re pleased to inform you that your manuscript has been judged scientifically suitable for publication and will be formally accepted for publication once it meets all outstanding technical requirements.

Kind regards,

Alberto Aleta

Academic Editor

PLOS ONE

---

## [Editor Report · Acceptance letter]

4 Aug 2023

PONE-D-22-32220R1 

Empirical distributions of time intervals between COVID-19 cases and more severe outcomes in Scotland 

Dear Dr. Kao:

I'm pleased to inform you that your manuscript has been deemed suitable for publication in PLOS ONE. Congratulations! Your manuscript is now with our production department. 

Kind regards, 

on behalf of

Dr. Alberto Aleta 

Academic Editor

PLOS ONE